
# 1  Chronicle of a forecast flood: exposure and vulnerability
# 2  on the southeast coast of Spain

Rubén Giménez-García[1], Víctor Ruiz-Álvarez [1], Ramón García-Marín[1]
[1]Department of Geography, University of Murcia, 30001 Murcia, Spain.
*Correspondence to:* Rubén Giménez García (ruben.gimenez@um.es)
**Abstract.** In recent years, flooding has become one of the main natural disasters which poses the greatest
risk, impact and bearing on the coastal areas of southeast peninsular Spain. Traditionally, the analysis of
torrential precipitation events that cause floods has been predominant in the assessment of the factors that
lead to this type of catastrophe. At present, despite considering this meteorological phenomenon as the
trigger of river overflows, responsibility corresponding to the human factor in the perpetration of the risk
of a natural disaster should not be forgotten. This study will ascertain the influence of urban and real
estate development in increasing exposure to floods. In this sense, the pluviometric observations obtained
in different precipitation events have been counted, mapped and analysed. Likewise, the evolution and
development of the real estate portfolio is examined. The information obtained has been crossed with the
digital cartography of flooded areas (National System of Flood Zones Cartography, SNCZI in spanish
acronym). The main finding of the study shows that there seems to be unquestionable evidence to
attribute a good part of the existing risk to the territorial transformation and to the continuous process of
artificialization of the soil recently carried out.

## 19  1. Introduction

In recent decades the Mediterranean coast has experienced an increase in vulnerability and exposure to
the danger of torrential rains (Olcina, 2017; López et al., 2017, Ribas et al., 2020). Consequently, there
has been an increase in economic losses related to floods associated with rainstorms (Marchi et al., 2010;
Spekkers et al., 2013; López, 2019). One of the main causes of extraordinary risk insurance cover in
Spain is flooding. It occupies first place when measured against other causes in the total cost of
compensation. Flooding represents 72.3% of the total over the period 1987-2019 (Insurance
Compensation Consortium, 2019).
The significant increase in the number of floods and their consequent damage to the coastal belt,
according to precipitation records, is related to a multiplication of exposure and vulnerability to these
events, rather than an increase in extreme rainfall phenomena (López et al., 2017; Pérez et al., 2018).
Regionalized climate change projections do not show a significant increase in the frequency and
magnitude of floods in the Mediterranean regions (Kundzewich et al., 2006; Rajczak et al., 2013; Madsen
et al., 2014; Alfieri et al., 2015). Furthermore, according to the IPCC (2018), there is little confidence that
anthropogenic climate change increases the frequency of floods on a global scale.
On the other hand, over the last decades there has been an intense urbanization process on the Spanish
Mediterranean coast, especially between 1997 and 2007, which experienced the greatest real estate boom
(Burriel, 2008). The great expansion experienced by new residential areas and agricultural crops has led
to the improper occupation of the fluvial channels and floodplains (Rico et al., 2010; Gil, 2014; Ibarra et
al., 2017; García, 2018). The development of new residential areas has led to a progressive increase in the
arrival of tourists, consequently, there is a greater exposure of the population to torrential rain events
(Meseguer et al., 2021).




This study aims to examine the causes that have led two coastal municipalities in the Region of Murcia
(Los Alcázares and San Javier) to become the towns which are most affected by flood processes. To this
end, the intensity of these precipitation events that have acted as triggers of the floods is evaluated. The
analysis of these rainfall episodes makes it possible to assess any responsibility that corresponds to the
natural factor in the development of the catastrophe.
Once the vigour and regularity with which torrential precipitation events occur, the implication of the
human factor in increasing vulnerability to the risk of flooding is examined. For this, the building
propagation process and the degree of exposure that the real estate portfolio presents to the possibility of
suffering flood events in different return periods is analysed. The comparison of geo-referenced land
registry data corresponding to the various construction phases and impulses, together with the amount and
disposition of land threatened by flooding (at different times of occurrence), which allows an estimation
of the implication that territorial planning and management have on the perpetration of these natural
disasters.
**2. Methodology**
**2.1. Study area**
The territorial scope of this research study is located in the coastal area of the Region of Murcia
(Southeast of the Iberian Peninsula). Unlike most municipalities located on the Mediterranean coastline,
the administrative boundaries being studied are surrounded by the brackish lagoon of the Mar Menor
(Fig. 1). There are two towns whose spatial extension is as follows: 2,005.2 hectares in Los Alcázares,
and 7,518.04 hectares in San Javier. Both municipalities are located in the watersheds of the Albujón and
Maraña Ramblas (ephemeral channels), which together comprise some 104,800 hectares. This territorial
area exhibits a semi-arid climate, with an average annual rainfall of 300 mm. Although these
precipitations show a notable torrential nature (Castejón et al., 2017).
The territorial peculiarities (high insolation and thermal bonanza, among others) which characterize this
territory have boosted its traditional consideration as an outstanding tourist and second-home destination,
especially within the national tourist scene. The boom in this sector of activity has caused an increase in
the demand for accommodation (tourist and residential), which has had a significant impact on an intense
building process that has filled a large part of its territory, including areas less suitable for the
construction of dwellings, such as the alluvial plains that make up the aforementioned ephemeral
channels. The improper execution of urban processes in these sectors has increased the exposure and
vulnerability of the territory under analysis, creating residential areas with a high risk of flooding.

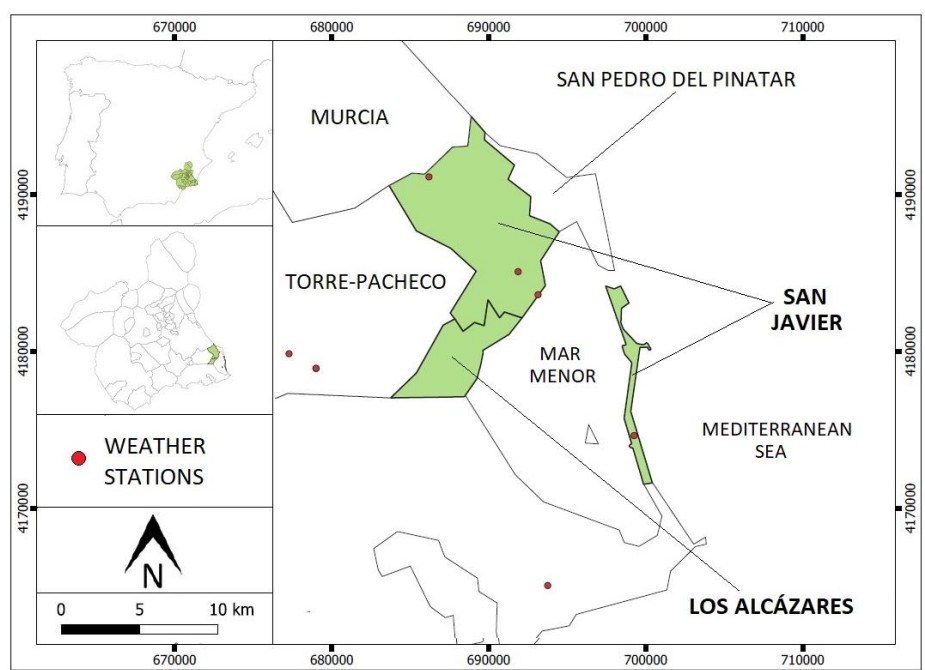


**Figure 1: Location map of municipalities analysed in the Region of Murcia (Los Alcázares and San Javier).**

**Source: own elaboration.**

## 2.2. Material and Methods

The following analysis is based both on the combination of data obtained from different sources, as well as the use of various spatial component techniques that allow contrasting georeferenced information arranged on the earth's surface. In this sense, the pluviometric observations obtained in different precipitation events have been counted, mapped and analysed through the official meteorological stations of AEMET (State Meteorological Agency) or SIAM (Agricultural Information System of Murcia), as well as meteorological amateurs who are individuals made up of associations: Meteoclimatic and AMETSE (Southeast Meteorological Association). The precipitation data has been homogenized (Ruiz, 2020) and interpolated using the technique known as Ordinary Kriging (Vicente et al., 2003; Moral, 2010; Zucarelli et al., 2014).

Together with the analysis of the natural component (precipitation), the social implication in the evolution of the flooding processes will be ascertained. For this, the evolution and development of the real estate portfolio is studied through the use of alphanumeric data (represented graphically on the territory) provided by the General Directorate of Land Registry. This official body belonging to the Ministry of Finance (Government of Spain) discloses current and detailed information on the multitude of variables that characterize the national building complex. The information obtained from this database has been crossed with digital cartography of floodplains of fluvial origin propagated by the National System of Flooding Zones Cartography (Ministry for the Ecological Transition and the Demographic Challenge). The contrast of both variables allows the degree of vulnerability and the amount of buildings and constructed land area that remain exposed to possible flooding processes in different periods of return or occurrence to be studied.



The detailed analysis of this data set can reveal, in some way, the amount of responsibility that
corresponds to the natural factor (magnitude of episodes of torrential rains) and human factor
(management and territorial planning) in the development and increase of the risk raised due to a danger
of natural origin that has become a catastrophe in the areas examined in recent years.
**3. Results and argument**
**3.1. Pluviometric analysis**
Firstly, the evolution of the pluviometric episodes with a precipitation greater than 100mm (Fig. 2) that
occurred in the territorial scope of the slopes of the Albujón and Maraña ephemeral channels is shown.
The selected time slot is 1934-2020. During this period, a total of 51 episodes were recorded, the majority
during the autumn (28.55% of the total) and winter (16.30% of the total) months. Over the last decades, a
slight upward trend has been seen, as is the case in other parts of the Mediterranean coast (Olcina, 2017).
In the reference period 1961-1990, a total of 16 episodes of abundant rainfall were recorded, and in the
current reference period: 1991-2020, a total of 21 episodes. Simultaneously, there has also been an
increase in their magnitude. The greatest increase is observed in the winter months. Likewise, there is an
evident transfer of the month of greatest occurrence of these rainfall events from October to September.

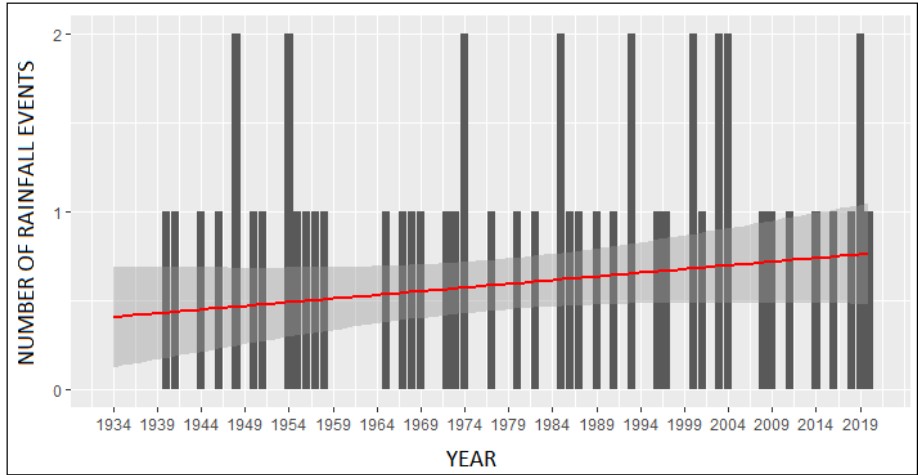

**Figure 2: Evolution of the number of precipitation episodes ≥ 100 mm. Source: Own elaboration, data**
**obtained from AEMET and SIAM.**
In the entire period analysed, a total of 4 episodes have occurred with a record greater than 200 mm. All
of them have been recorded since the 1970s: October 1972, November 1987, December 2016 and
September 2019. The last two recorded which occurred over the last five years are, without a doubt, the
episodes which have caused the greatest amount of damage and economic loss in the analysed territory
(Espín et al., 2017; García et al., 2020; Martí et al., 2021). In both episodes, 200 mm was exceeded in
large areas of the analysed territory, the most affected area being the mouths of the Albujón and Maraña
ephemeral channels, where the municipalities of San Javier and Los Alcázares are located. These two
rainfall events were characterized by high torrentiality and hourly intensity.
Fig. 3 shows the spatial distribution of the accumulated precipitation in the watersheds of the Albujón and
Maraña ephemeral channels, over the two mentioned episodes. The first of them, which occurred between
December 15th-19th, 2016, was caused by an intense east wind flow from a long maritime route favoured
by the strengthening of the Central European anticyclone (Espín et al., 2017). In the southeast of the
Spanish peninsula, losses were estimated at 67.4 million euros, with the municipality of Los Alcázares
being the most affected (Consorcio de Compensación de Seguros, 2019). The second episode, which
occurred between September 11th-15th, 2019, was caused by an Isolated Depression at High Levels
(DANA), which caused the formation of mesoscale convective systems (Martí et al., 2021). During this
event, the economic losses were much greater than in the 2016 episode, estimated at 479 million euros
(Consorcio de Compensación de Seguros, 2019). This is the fourth most serious event in terms of
compensation by the Insurance Compensation Consortium since 1971, just behind the 1983 floods in the
Cantabrian Sea (Pejenaute, 1991), the atypical cyclonic storm "Klaus" in 2009 (Velázquez, 2015) and the
2011 Lorca earthquake (Pina et al., 2015). Again, the municipality of Los Alcázares was the most
affected municipality, in this case together with the neighbouring town of Orihuela in Alicante (Nuñez,
136    2019).

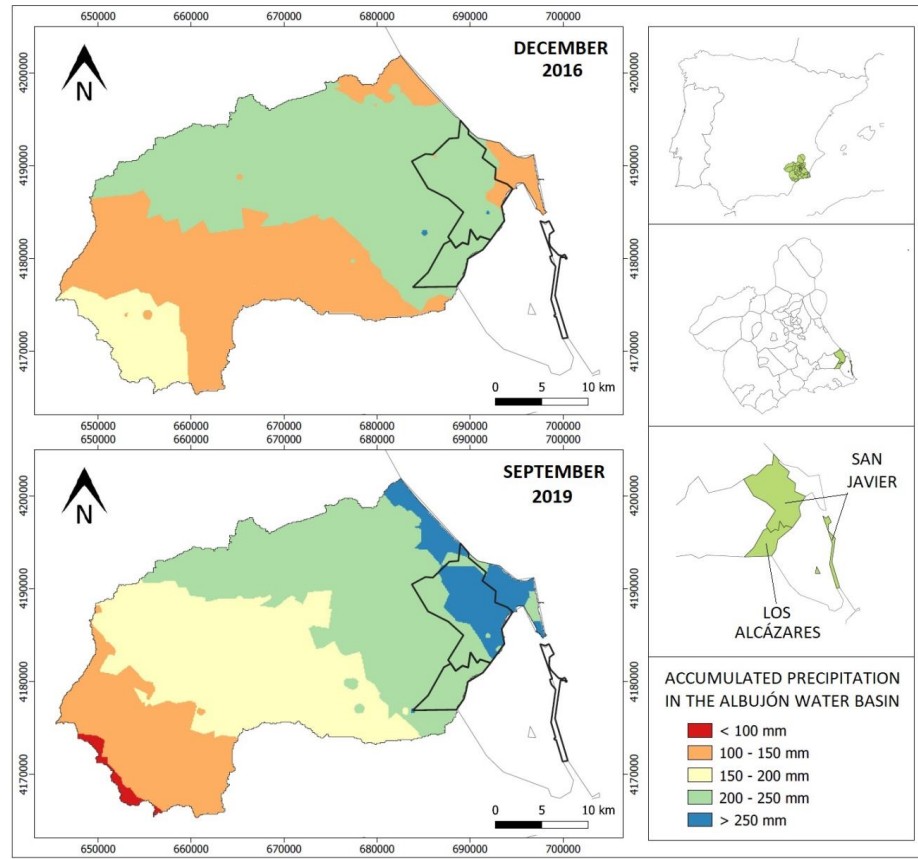

**Figure 3: Spatial distribution of accumulated precipitation during the events of December 2016 and**
**September 2019. Source: Own elaboration, data obtained from AEMET and SIAM.**





### 3.2 Land registry analysis: increase in buildings

Over the last few decades, the earth's surface has been subjected to an unprecedented artificialization process (Lois et al., 2016). This boom in anthropizing is motivated by building expansion, which has clearly proliferated on the Murcian coast (Pérez et al., 2018). The excessive urbanization phenomenon encourages vulnerability and increased exposure to the risk of flooding (Olcina, 2004).

The towns of Los Alcázares and San Javier have experienced a drawn-out increase in their real estate portfolio. Overall, the number of constructions registered by the General Directorate of Land Registry (in both locations) has gone from accounting for just over a thousand blocks of flats and houses built in 1950, to close to 16 thousand in 2019. The area occupied by these constructions has evolved parallel to the total number of buildings (Tab. 1).

| | | 1950 | 1980 | 2000 | 2019 |
|---|---|---|---|---|---|
| LOS ALCÁZARES | NUMBER OF BUILDINGS | 471 (44.56 %) | 1903 (28.45 %) | 4595 (33.82 %) | 5282 (33.08 %) |
| | SURFACE AREA OCCUPIED BY BUILDINGS (Ha.) | 6.81 (42.30 %) | 36.86 (27.41 %) | 111.88 (34.99 %) | 149.37 (33.19 %) |
| SAN JAVIER | NUMBER OFBUILDINGS | 586 (55.44 %) | 4785 (71.55 %) | 8990 (66.18 %) | 10687 (66.92 %) |
| | SURFACE AREA OCCUPIED BY BUILDINGS (Ha.) | 9.28 (57.70 %) | 97.64 (42.59 %) | 207.91 (66.18 %) | 300.69 (66.92 %) |
| TOTAL | NUMBER OF BUILDINGS | 1057 | 6688 | 13585 | 15969 |
| | SURFACE AREA OCCUPIED BY BUILDINGS (Ha.) | 16.09 | 134.50 | 319.79 | 450.06 |

**Table 1: Evolution of the number of buildings and built area. Source: Own elaboration, data obtained from the General Directorate of Land Registry.**

Since the middle of the 20th century, land obstructed by construction has increased by almost 2,700% (from 16.09 ha. to 450.06 ha.). This figure exceeds the percentage of development observed by the number of infrastructures erected (1,410%). Now, the construction development carried out individually by each municipality shows different rhythms and intensities. In this context, the number of buildings and land occupied in San Javier is, at all times, higher than that registered in Los Alcázares. In this regard, in the starting year of the series, San Javier had 586 buildings (55.44%) occupying 9.28 ha. (57.70%), values that exceed the figures published by Los Alcázares (471 buildings and 6.81 occupied hectares).

Over the years, the evolution of these indicators tends to increase, expanding the existing building gap between both local municipalities. Consequently, over the last seven decades, the number of recognized buildings in San Javier has increased by 1,723% (10,687 in 2019) and the area occupied by them by 3,137% (300.69 ha. in 2019). On the other hand, in the same period of time, the buildings and cemented land accounted for in Los Alcázares increased by 1,021% and 2,095% (5,282 constructions and 149.37 ha. in 2019), respectively (Fig. 4).
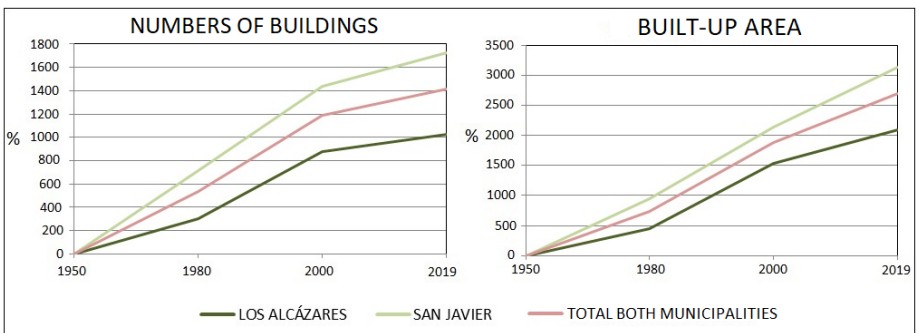

**Figure 4: Evolution of the number of buildings and built area (1950, 1980, 2000 and 2019). Source: Own elaboration, data obtained from the General Directorate of Land Registry.**

The cartography shown in Fig. 5 shows the vigour of the construction process registered in both municipal areas. The presence of real estate infrastructures in 1950 is scarce, with an urban area partially concentrated in Los Alcázares and a scattered settlement in San Javier. Two decades later, the top of Los Alcázares expands and a new settlement begins to sprout northeast of it, which is named Los Narejos. Meanwhile, San Javier consolidates its central nucleus and experiences the development of a large residential area (Santiago de La Rivera), which sits on the coastline located to the east of the main urban area, the so-called real estate bubble (Fernández, 2016; Jerez et al. 2012). The rise in tourism has caused the urban complexes located on the Mediterranean coast at the beginning of the 2000s (and Mar Menor) (Serrano, 2007; López and Pérez, 2017), driven by the expansion of traditional population centres and the development of "resort" -type residential complexes with recreational, leisure and entertainment spaces (golf courses) (François, 2010).

From the beginning of the 21st century to the present, the urban phenomenon has undergone two phases clearly marked and influenced by the economic context the country is going through. Until 2008, the national complex and, fundamentally, the coastal areas, saw one of the most notable building drives in their history. However, from then until now, the bursting of the real estate bubble encouraged by the economic recession has caused the stagnation of construction development (Bernardos, 2009; Górgolas, 2019). In addition, the spatial field that our research focuses on has been heavily weighed down by the environmental problems that arose around the Mar Menor and the impact caused by successive extreme meteorological events and related floods (León et al., 2017).

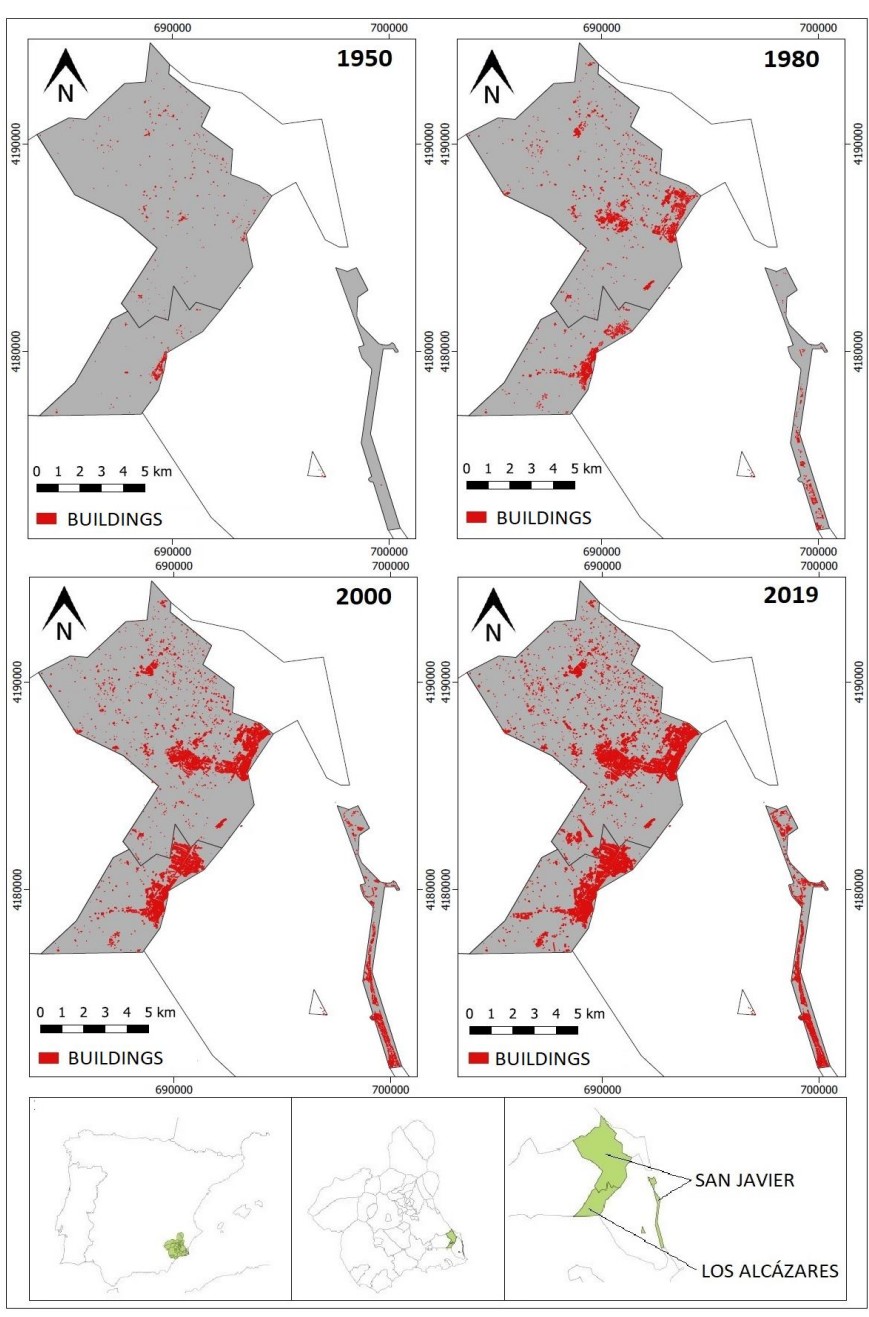

**Figure 5: Mapping of the building development in Los Alcázares and San Javier. Source: Own elaboration, data obtained from the General Directorate of Land Registry.**

If the evolution of real estate and the area occupied by buildings is analysed according to the established time periods, the relevance of the construction process carried out over the last two decades of the 20th century can be verified. The building dynamics carried out between the years 1980 and 2000 is of great





magnitude with more than 210 properties being built a year in San Javier and about 135 in Los Alcázares.
The area occupied by these buildings, in the same period of time, stands out above the amount of land
obstructed by constructions in the rest of the temporary spaces (5.51 ha. in San Javier and 3.75 ha. in Los
Alcázares). Nevertheless, between 2000 and 2019, despite constituting the phase with the lowest average
annual number of constructions (89 in San Javier and 36 in Los Alcázares), the proposed building
typology (mainly single-family homes) occupies a larger surface (Fig. 6).

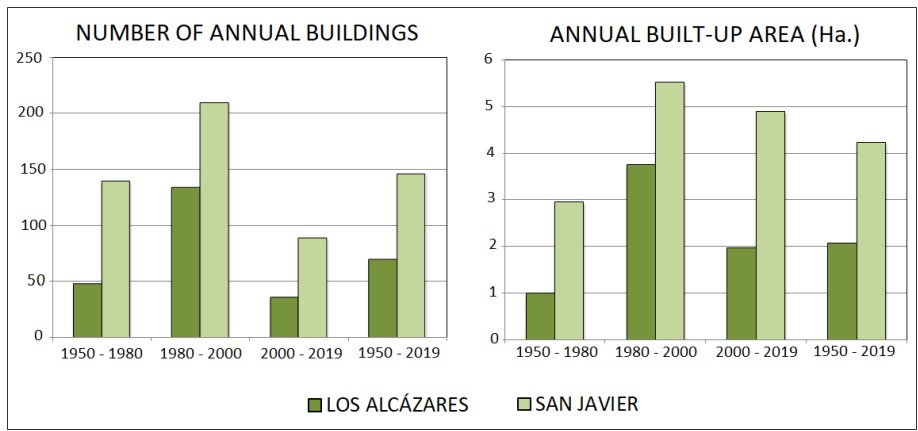

**Figure 6: Number of constructions and constructed area annually over time (1950 - 1980, 1980 - 2000, 2000 -**
**2019 and 1950 - 2019). Source: Own elaboration, data obtained from the General Directorate of Land**
**Registry.**
The average number of completed properties annually in the series as a whole (146 in San Javier and 70
in Los Alcázares) was only exceeded in the 1980-2000 period. The number of buildings erected between
1950 and 1980 is also higher than those between 2000 and 2019. In spite of this, due to their placement in
terms of height, the ground occupied by the finalised infrastructure between the first dates has caused
them to spread over less surface area than those recently outlined.
**3.3. Areas of exposure**
Once the dynamics of real estate production have been analysed, the exposure of buildings built in
different time periods to events of river overflow will be estimated. To do this, the land registry data
shown above is contrasted with territorial information on areas at risk of flooding obtained from the
National System of Flood Zones Cartography (SNCZI). This project developed by the Ministry for the
Ecological Transition and the Demographic Challenge of Spain (MITECO) provides georeferenced
documentation on the probability of a flood in a territory in different periods of time (López, 2020). In
this respect, it differs between areas with high risk of flooding (return period of 10 years), frequent (return
period of 50 years), medium or occasional (return period of 100 years) and low or exceptional (return
period of 500 years).
Among these four river overflow scenarios, this study focuses its attention on the most extreme episodes,
trying to investigate the possible effect caused on the housing portfolio by a meteorological event in the
return periods (RP) of 10 and 500 years. As is evident, the reason that supports the analysis of the first
context of occurrence lies in the fact that it is the most likely situation (RP 10 years). For its part, the
choice of the most anomalous return period (500 years) incurs that, as can be seen in Fig. 7, the last


precipitation event that occurred in the area being studied (September 2019) exceeded the established
perspectives by the state agency in charge of delimiting the Hydraulic Public Domain (DPH) areas in its
studies.

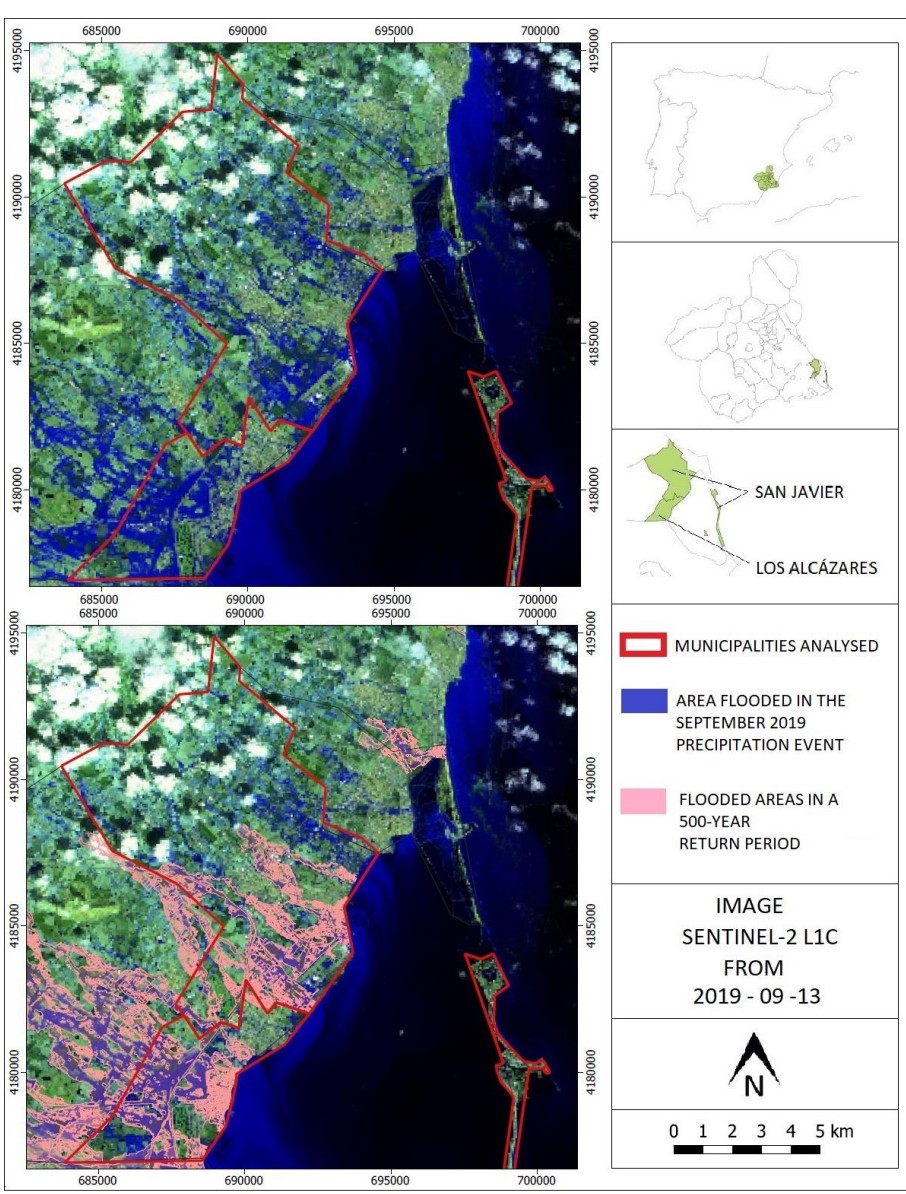


**Figure 7: Satellite image of the flooded area after the torrential rainfall event in the territory under analysis**
**(September 2019). Source: Own elaboration, data obtained from Sentinel and SNCZI.**






**A) Return period of 10 years (RP10)**
The flood scenario with the highest occurrence level reveals the probability of river overflows occurring
once every ten years, that is, a 10% contingency. The effect of this return period on buildings is much
more contained than that of 500 years, logically. However, its impact on the housing stock of the
municipalities analysed in this study shows a level of affectation to take into account (Tab.2).

240     .

| | | 1950 | 1980 | 2000 | 2019 |
|---|---|---|---|---|---|
| **LOS ALCÁZARES** | NUMBER OF EXPOSED BUILDINGS | 287 (60.93%) | 889 (46.72%) | 2035 (44.29%) | 2230 (42.22%) |
| | SURFACE AREA OCCUPIED BY EXPOSED BUILDINGS (Ha.) | 3.56 (52.25%) | 15.78 (42.81%) | 32.78 (29.30%) | 38.49 (25.77%) |
| **SAN JAVIER** | NUMBER OF EXPOSED BUILDINGS | 54 (9.22%) | 631 (13.19%) | 1282 (14.26%) | 1610 (15.07%) |
| | SURFACE AREA OCCUPIED BY EXPOSED BUILDINGS (Ha.) | 0.69 (7.44%) | 1.41 (11.44%) | 31.26 (15.04%) | 46.79 (15.56%) |
| **TOTAL** | NUMBER OF EXPOSED BUILDINGS | 341 (32.26%) | 1520 (22.73%) | 3317 (24.42%) | 3840 (24.04%) |
| | SURFACE AREA OCCUPIED BY EXPOSED BUILDINGS (Ha.) | 4.25 (26.40%) | 17.19 (12.78%) | 64.04 (20.03%) | 85.28 (18.95%) |

**Table 2: Absolute evolution and percentage representation of the number of buildings and surface occupied by**
**them exposed to flooding in a RP of 10 years Source: Own elaboration, data obtained from the General**
**Directorate of Land Registry and SNCZI.**
Despite San Javier having undergone a more organised urbanization process than the municipality of Los
Alcázares (as reflected in the previous section), the exposure to the ephemeral channels presented by the
group of buildings in the latter town is notably higher. The evolution of the percentage representation of
buildings and surface area occupied by them, in relation to the total amount of municipal buildings in
both administrative boundaries, follows opposite trends. In this regard, while the most affected locality
(Los Alcázares) sees the exposure of its housing blocks decrease in percentage terms, San Javier reveals
an increase in vulnerability due to greater exposure, since between 1950 and 2019 the affected area
doubled. On the contrary, between the beginning and the end of the series, the percentage of exposed
land in Los Alcázares is reduced by half. Nevertheless, the territorial area affected by floods in a return
period of 10 years in this last locality (Los Alcázares) increased notably, going from 3.56 ha. in 1950 to
38.49 ha. in 2019. An explanation for this contradiction lies in the fact that not all the urban expansion
process carried out in the municipalities is located in areas with a probability of flooding every 10 years,
as is the case of Los Narejos (Fig. 8).
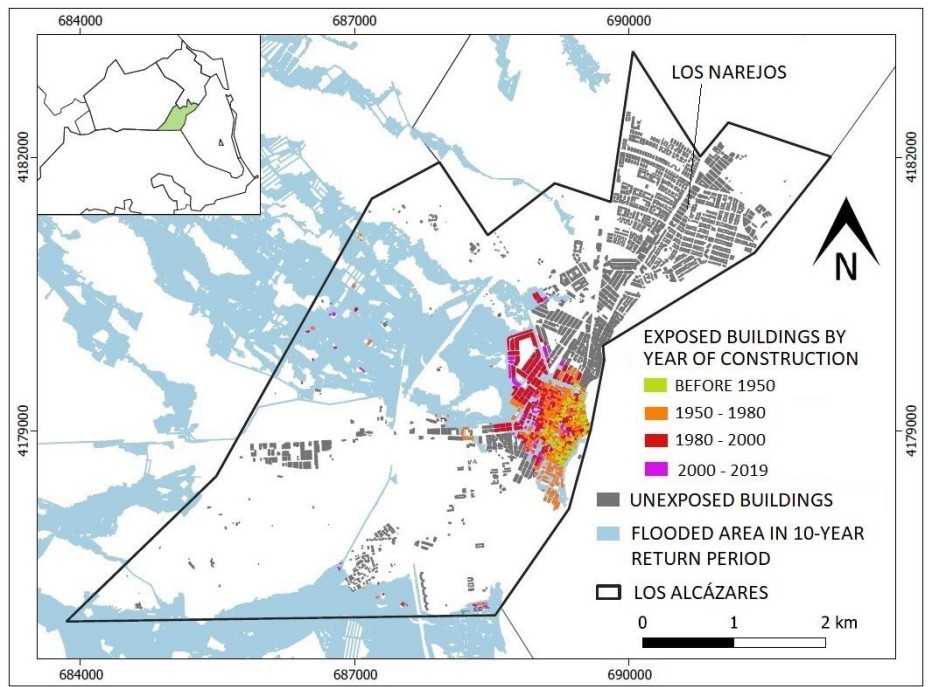

**Figure 8: Buildings exposed to floods in a 10-year RP (Los Alcázares). Source: Own elaboration, data obtained from the General Directorate of Land Registry and SNCZI.**

In the same way, the percentage representation of total buildings affected by this return period drops from 60.93% to 42.22%. Nonetheless, the absolute value reveals that the number of buildings exposed to a flood every ten years goes from 28 buildings in the middle of the 20th century, to about 2,230 at the end of the second decade of the 21st century. At the beginning of the new millennium, this number exceeds two thousand buildings with a high risk of flooding. Something similar, although at a lesser level, takes place in San Javier, a municipality in which 1,282 buildings exposed to river overflows were counted in 2000. This is a figure that in the total computation of the series has increased by more than 1,500 buildings.

Altogether, 24.04% (3,840) of the properties and 18.95% (85.28) of the territory occupied by them in 2019 may present a risk of flooding at least once every 10 years. The percentage of these buildings and land area occupied in 2019 has decreased considerably since the first year it was observed (Fig. 9).

It is interesting to mention that when focusing the research on floods of fluvial origin, the territorial spit corresponding to La Manga del Mar Menor (included in the municipality of San Javier) is not to be considered due to the lack of surface hydrographic currents.

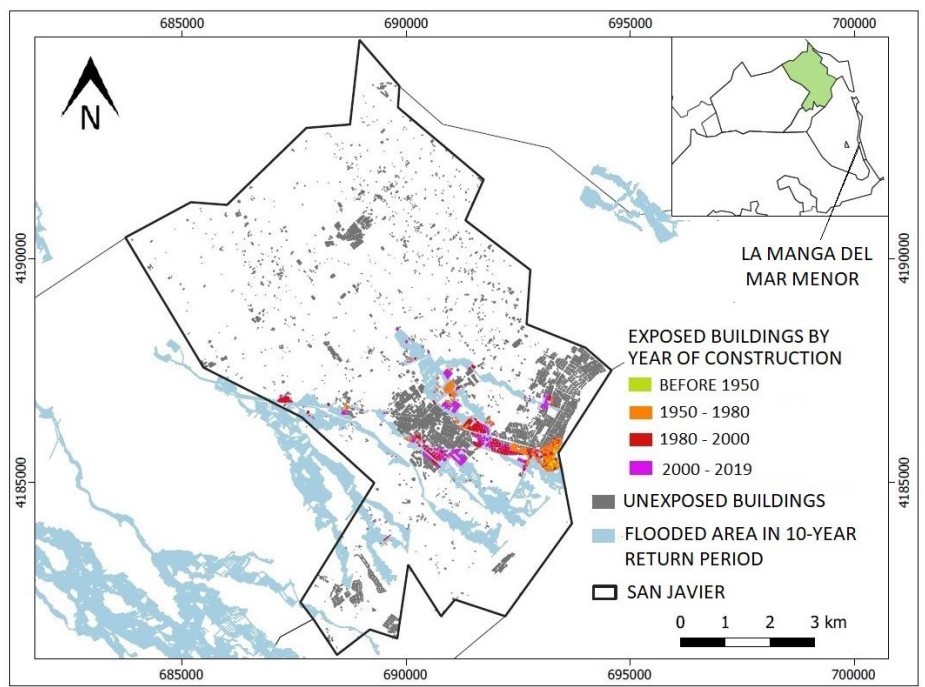

**Figure 9: Buildings exposed to floods in a 10-year RP (San Javier). Source: Own elaboration, data obtained from the General Directorate of Land Registry and SNCZI.**

If the existing exposure factor between time periods is analysed, it is possible to see the number of buildings and the land occupied in areas at risk of flooding during different construction cycles. To this end, during the time series examined, in the two municipalities as a whole, 3,499 buildings were erected with a very high risk of flooding (RP 10 years), which represents 23.46% of all buildings built in this area within the temporary parenthesis. Los Alcázares stands out, with more than 40% of the buildings built in the last seven decades (4,811), with a very high probability of flooding (Fig. 10). This threshold of representation of properties exposed to ephemeral channels is also exceeded, by this same municipality, in the building phases 1950-1980 (620) and 1980-2000 (1,146).

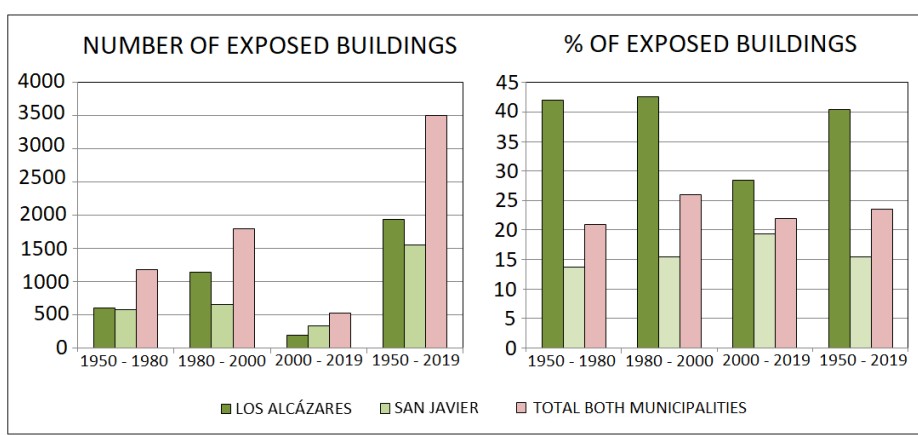

**Figure 10: Number and percentage representation of buildings exposed to flooding in a 10-year RP by**
**temporal phases. Source: Own elaboration, data obtained from the General Directorate of Land Registry and**
**SNCZI.**
On the other hand, the percentage of surface exposed to this phenomenon with meteorological origin only
exceeds 40% (12.23 ha.) Of all the land occupied in the first urban development cycle (1950-1980). The
amount of properties that show vulnerability in San Javier during this initial phase is very small, without
reaching one hectare (0.81% of everything built). On the other hand, in the two subsequent temporal
phases (1980-2000 and 2000-2019), San Javier has a percentage of constructed surface area in evident
danger of flooding higher than both the municipality of Los Alcázares and the aggregate of the two
localities (Fig. 11).

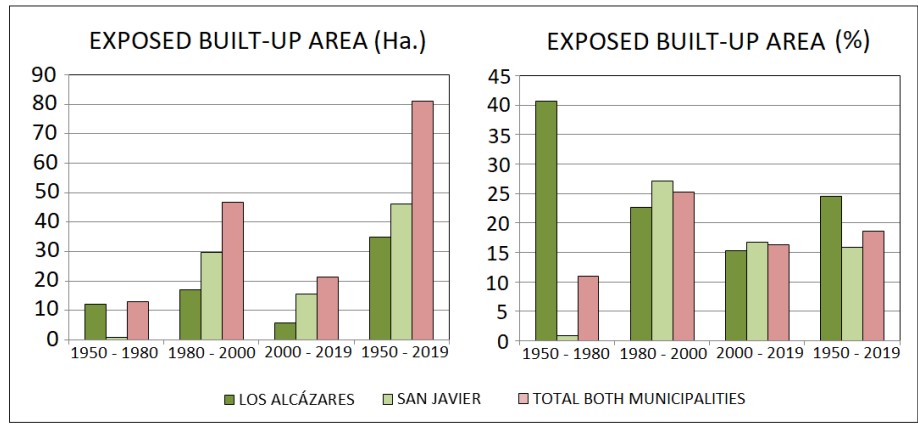


**Figure 11: Number and percentage representation of constructed area exposed to flooding in a 10-year RP by**
**temporal phases. Source: Own elaboration, data obtained from the General Directorate of Land Registry and**
**SNCZI.**
This representation of land built in San Javier and at risk of flooding once every decade shows greater
strength in the area occupied between 1980 and 2000 (more than a quarter of the built area). In addition, it
is the period of time in which the largest number of hectares with high flood risk is built in each of the
spatial areas examined.
**B) Return period of 500 years (RP500)**
The impact exerted by floods in the period of least probable occurrence (RP 500 years) is much more
prominent than in the one previously analysed. In this sense, the amount (absolute and percentage) of
buildings exposed to the danger of this phenomenon is very high.
As in the case of 10-year-old RP, the influence exerted by the Albujón ephemeral channel in Los
Alcázares increases the vulnerability of its housing portfolio. In this regard, 9 out of 10 buildings built in
this municipality before 1950 may be affected by an ephemeral channel. The excessive representation of
buildings exhibited until the middle of the last century tends to decrease as we approach the present day.
Now, in 2019, the percentage of buildings and surface exposed to ephemeral channels for a RP of 500
years remains very high, comfortably affecting more than half of the real estate portfolio (Tab. 3). This
value almost doubles that shown by the group of localities and triples that revealed by San Javier (around
22% of buildings and exposed area).





|  |  | 1950 | 1980 | 2000 | 2019 |
|---|---|---|---|---|---|
| LOS ALCÁZARES | NUMBER OF EXPOSED BUILDINGS | 421 (89.38%) | 1460 (76.72%) | 3006 (65.42%) | 3493 (66.13%) |
|  | SURFACE AREA OCCUPIED BY EXPOSED BUILDINGS (Ha.) | 5.90 (86.70%) | 27.44 (74.43%) | 63.77 (56.99%) | 89.07 (59.63%) |
| SAN JAVIER | NUMBER OF EXPOSED BUILDINGS | 96 (16.38%) | 1010 (21.11%) | 2017 (22.44%) | 2448 (22.91%) |
|  | SURFACE AREA OCCUPIED BY EXPOSED BUILDINGS (Ha.) | 1.41 (15.17%) | 22.72 (23.27%) | 44.81 (21.55%) | 66.98 (22.28%) |
| TOTAL | NUMBER OF EXPOSED BUILDINGS | 517 (48.91%) | 2470 (36.93%) | 5023 (36.97%) | 5941 (37.20%) |
|  | SURFACE AREA OCCUPIED BY EXPOSED BUILDINGS (Ha.) | 7.31 (45.43%) | 50.16 (37.29%) | 108.58 (33.95%) | 156.05 (34.67%) |

**Table 3: Absolute evolution and percentage representation of the number of buildings and surface occupied by**
**them exposed to flooding in a RP of 500 years. Source: Own elaboration, data obtained from the General**
**Directorate of Land Registry and SNCZI.**
The town located to the north of the study area has traditionally remained less exposed to floods. Despite
this, the vulnerability exhibited by the exposure factor of its buildings and the surface area that supports
them has increased over the years. As of today (2019), San Javier has 2,448 buildings and 66.98 ha which
are exposed to this phenomenon.
As can be seen in the cartography referring to Fig. 12, the southern strip of the administrative delimitation
of San Javier is the area most affected by river overflows. Coincidentally, it is also the municipal area that
is least clogged by urban development, which is why the impact of the floods is not so prominent.

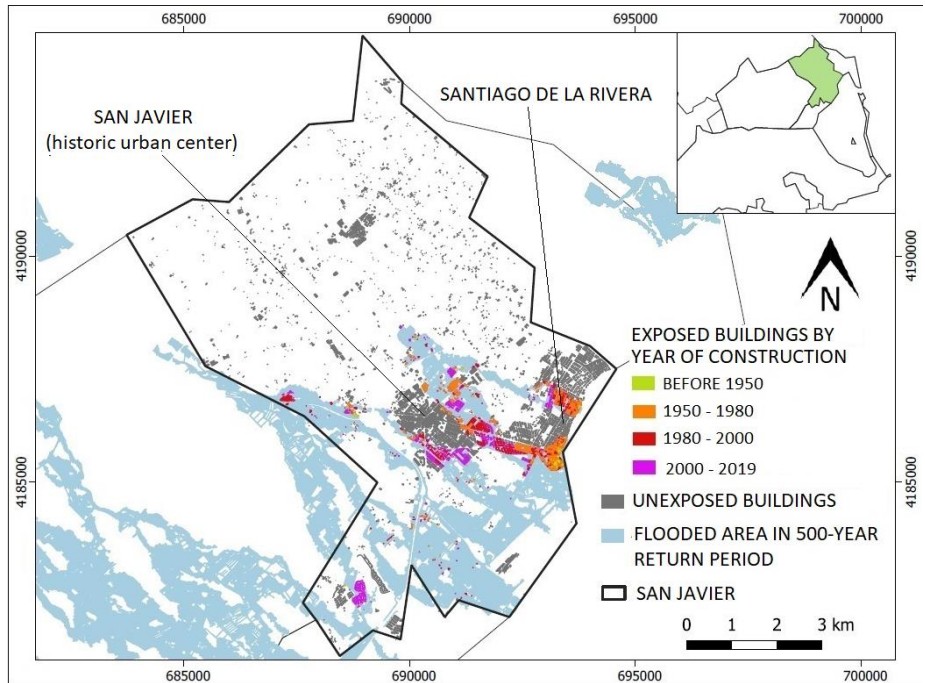


**Figure 12: Buildings exposed to floods in a 500-year-old RP (San Javier). Source: Own elaboration, data**
**obtained from the General Directorate of Land Registry and SNCZI.**

The northeast area of Santiago de La Rivera and the historic urban centre of San Javier are practically not
affected by the water level. However, the most recent constructions, located in the peripheral sector of the
headwaters and areas that join the two urban centres, have greater vulnerability. This fact means that,
unlike Los Alcázares, the most recent building developments have more exposure. In this sense, Fig. 13
shows how the central area of Los Alcázares is the area with the highest risk of flooding in a period of
occurrence of 500 years.



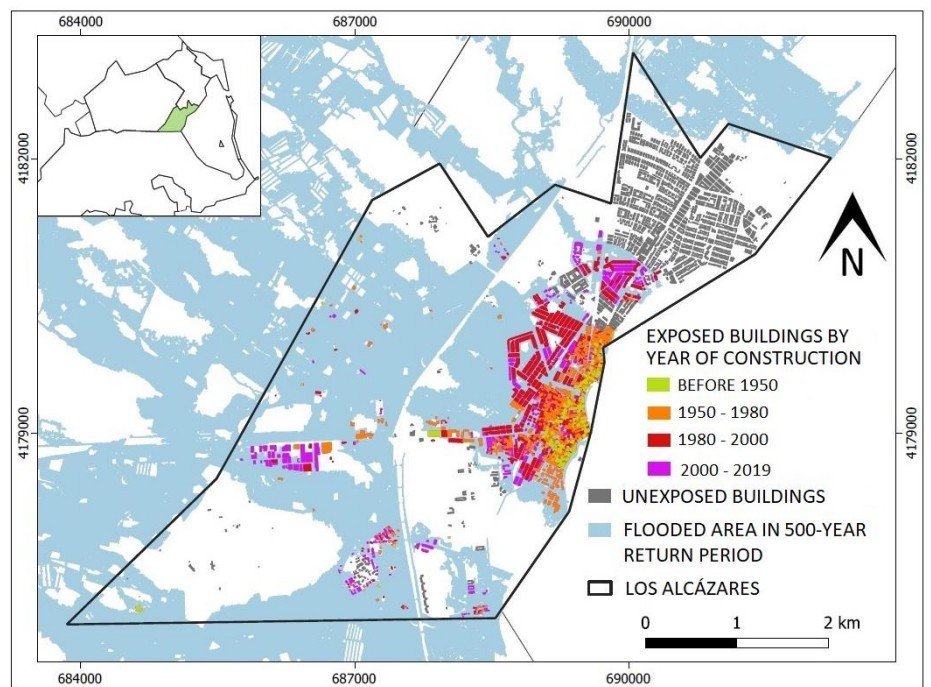

**Figure 13: Buildings exposed to floods in a RP of 500 years (Los Alcázares). Source: Own elaboration, data**
**obtained from the General Directorate of Land Registry and SNCZI.**
The evaluation of properties and surface area affected by floods in a period of occurrence of 500 years
shows values which should be highlighted. During the development of the complete series, the amount of
infrastructure and land exposed by both municipalities amounts to 5,424 properties and 148.74 ha. As is
usual in this analysis, the possible impact of this phenomenon is more telling in Los Alcázares than in San
Javier, with 56.64% (3,072) of these buildings belonging to the first mentioned location. The percentage
of possible damage to buildings erected in the last 70 years in Los Alcázares exceeds 63%. This
representation is exceeded in the periods 1950-1980 and 2000-2019, with percentage values that exceed
70% of the entire real estate portfolio (Fig. 14).

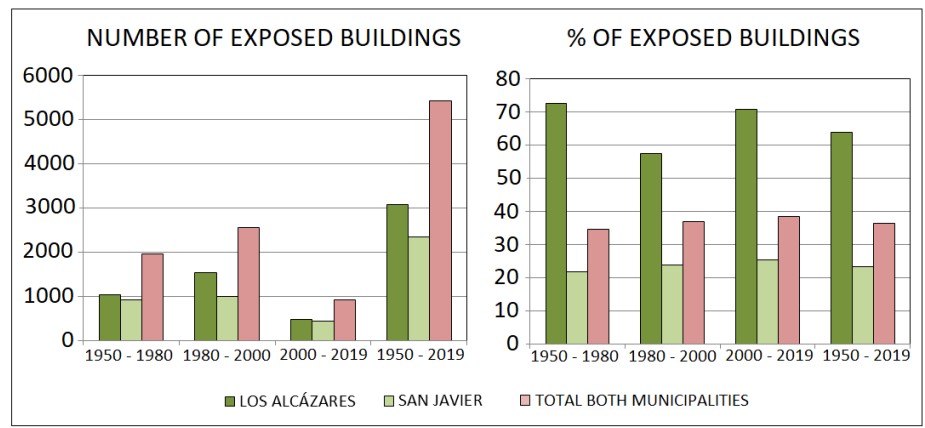

**Figure 14: Number and percentage representation of constructions exposed to flooding in a RP of 500 years by**


**temporal phases. Source: Own elaboration, data obtained from the General Directorate of Land**
**Registry and SNCZI.**
Regarding the number of buildings exposed to floods in a RP of 500 years in San Javier, it can be noted
that they are much more contained in each and every one of the established time phases. Thus, the 1980-
2000 construction cycle is the only one that registers more than a thousand exposed buildings. Unlike Los
Alcázares, the number of total properties likely to suffer the effects of a flood with this degree of
occurrence (500 years) barely exceeds 20% in each of the time periods analysed.
The percentage values of exposed land are practically similar to those observed in the analysis of
buildings. Nevertheless, and unlike these, the representation of occupied land in the 1980-2000 time
period in Los Alcázares is the only one that does not reach 50% (Fig. 15). Even so, it is the time period
with the highest absolute number of exposed surface area (36.33 ha.).

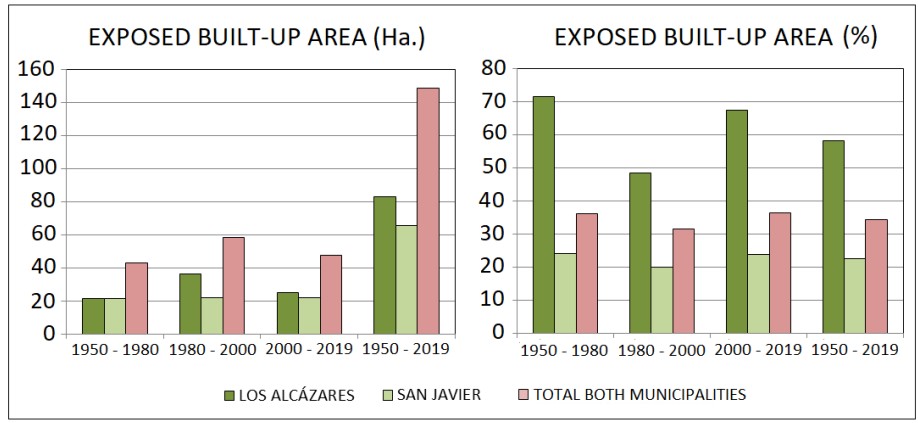

**Figure 15: No. and percentage representation of built area exposed to floods in a RP of 500 years by temporal**
**phases. Source: Own elaboration, data obtained from the General Directorate of Land Registry and SNCZI.**
More than a third of all the land occupied by buildings in the two municipalities, during each of the
periods studied, remains exposed. The total area of constructed land with the possibility of being affected
by a flood every 500 years is around 150 ha.
San Javier exhibits, in all the time cycles, an area at risk of flooding slightly higher than 20 ha, which
translates into a percentage value of around 20% of the entire built-up area.
Finally, despite the fact that both municipalities record the total number of buildings and hectares exposed
to flooding fairly evenly, the reduced total number of buildings and surface area occupied by them in Los
Alcázares (in relation to San Javier) make the percentage values of buildings and land (on which they sit)
at risk of flooding much higher than those exhibited in San Javier.
**4. Conclusions**
The flood processes, despite not having the social impact that they currently deserve, make up one of the
natural disasters with the greatest bearing and historical impact in the southeast of the Spanish peninsula.
The physical and morphological particularities of the water network that irrigates the analysed territorial
surface area, together with the meteorological irregularity that characterizes the Mediterranean climate,


stimulates the alternation of long periods of low water with punctual and vigorous torrential precipitation
events and related fluvial floods.
The occurrence of these floods is becoming increasingly serious, sometimes involving the loss of human
life and considerable material damage. Against this background, two positions persist in Spain that seek
to estimate the main cause that motivates the origin of these disasters. The first one, of technical-
administrative orientation, attributes the origin of the floods to the regularity and greater frequency of
torrential rain events. For its part, the ethical-geographical defence states that the risk and exposure to
floods has been gradually increasing as a consequence of the progressive transformation of territorial
coverage, the sealing of land and the unplanned proliferation of urban developments built in recent
decades.
After analysing both points of view, it is inevitable to highlight that both the natural phenomenon
(torrential rains) and the social factor (improper construction in flood areas) contribute to the risk of
flooding becoming a natural disaster. The two meteorological episodes analysed have prominent rainfall
records. In this sense, the intense precipitation events analysed force them to be considered as triggers of
the overflowing of ephemeral channels, flooding and flash floods experienced. However, despite
considering this ingredient as the essential and obvious origin of the phenomenon of flooding that
occurred, the responsibility that corresponds to the human factor in the perpetration of these catastrophes
should not be set aside. In fact, it is the human factor that can be corrected. In this way, it is undeniable to
attribute a large part of the existing risk to the territorial transformation and to the continuous process of
artificialization of the soil recently carried out.
The study carried out reveals how, in order to cover the growing residential needs of the tourist boom in
coastal areas, San Javier is disproportionately increasing its real estate portfolio, spreading urban
development over areas at obvious risk of flooding, which increases exposure and vulnerability of the
infrastructure and population.
For its part, despite having urban development in a less dynamic way, the municipality of Los Alcázares
has suffered the effects of the transformation of land uses and coverage carried out upstream of the fluvial
channel in which it is integrated, which increased significantly increased the risk of flooding. Thus, poor
management and planning in agricultural development has stimulated an increase in surface runoff, with
the diversion of river channels, lowering of plots, change of orientation of ephemeral channels and
waterproofing of soil. The result of these actions means that, unlike years ago, every time there is a
torrential rainfall event (not forgetting that torrential rains are typical of this territory and the
Mediterranean climate) the water network drains with greater force, overflowing the river and flooding,
fundamentally, the central area of the municipality.
Finally, it has been shown that despite being two examples or case studies with different urban
developments or spatial transformations, the effect or social impact is noted in the increase in exposure
and vulnerability related to flooding of rivers. As a result, territorial planning and management is
considered to be a fundamental and effective tool when trying to minimize or reduce the risk of flooding
and its disastrous consequences.



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
