# Peer review of "Chronicle of a forecast flood: exposure and vulnerability on the southeast coast of Spain"

_Natural Hazards and Earth System Sciences, 2021_

## Referee Comment (RC1)

Dear Editor, dear authors,

Thank you for the opportunity to review the manuscript "Chronicle of a forecast flood: exposure and vulnerability on the south-east coast of Spain". The manuscript analyses a relevant topic and of wide interest in the scientific community and would be worth of publishing in NHESS.

Nevertheless, there are some issues that need to be addressed before, being the first and foremost the sentence structure and punctuation, that is sometimes difficult to understand, being the abstract a clear example. The result is a bit confusing here and there. Maybe a final proof-reading by a native English speaker could be helpful.

On the other hand, the references list needs to be improved as there are a large number of missing references, listed in the text but not at the final list (page 3, Ruiz, 2020; Zuccarelli et al, 2014; page 5, page 7 as well and so on). Furthermore, there are references with a different citation or year of publication, as Lopez 2020 in page 9 when afterwards is cited as Lopez 2019. Please check carefully the list and correct all the mistakes.

Moving into specific comments, please see below:

**Introduction.**

You start writing about the Mediterranean coast being affected by torrential rains and an increase in vulnerability but the references used only refer to the Spanish Mediterranean coast. I suggest to add Spanish in line 20 or include references that show similar cases around the Mediterranean sea. There is a large number of works explaining similar events in France and Italy.

On line 27 you wrote "damage to the coast belt", again what coast belt are we talking about? Spanish? Mediterranean? A clarification would be welcomed.

Line 28, what do you mean with "according to precipitation records"? What is the relation between the floods you are talking about and precipitation. If we consider that floods within the Mediterranean basin are largely the result of extreme rainfall events then you should explain it clearly but, as written before, a careful English review should address such sentence.

Lines 36-40. Again confusing statements: "The great expansion….new residential areas and agricultural crops...led to the improper occupation of..." Can you provide with examples of such expansion in terms of land change. The you wrote "a progressive increase in the arrival of tourists" as a result of an increase of residential areas. The tourists were coming before, weren't they?. The large number of tourist is due to new residential areas or the new residential areas are built because of the arrival of a large amount of tourists? Furthermore, looks like only urban areas are subject to flooding and damages. Croplands are also flooded and large damages can arise from such events, so maybe some explaining should be included here.

Line 42: "towns which are the most affected to flood processes" compared to what? The whole region? Spain? A clarification would be needed.

Line 45 what do you mean with "catastrophe"? A flood is considered a catastrophe? Or human occupation of flood-prone areas can create a catastrophe? Please consider rewriting it.

Line 46 to 53. The whole paragraph is difficult to understand. Maybe some rewriting is needed again or include some parts in section 2.2

**Methodology**

Line 57. Maybe an explanation of the importance of being located along the Mar Menor could be included.

Line 59. "There are two towns" There are only two towns in the area or are they more? Why you choose those two? Please justify clearly your selection.

Line 63. What do you mean with torrential nature of precipitations? Explain how is the rainfall in the area and why torrential rain plays a large part in the development of floods.

Section 2.2 is somewhat confusing. Maybe a table including the data used and its sources should help the reader to understand the large work done to gather information.

Moreover, when you refer to the mixing of information from different sources (lines 90-95) you could add a figure showing it, again resulting in a clearer picture for the reader of your paper.

**Results and argument**

First at all, the word "argument" here is not clear. Do you mean "discussion"?

Line 102, why 100 mm? Justify the selection of such threshold as there is a large amount of published research about the topic. Moreover, such decision should not be included in section 2 and explained as a part of your research methods?

Line 104, the time period between 1934 and 2020 should be justified. Why those years?

Line 107, reference period. Again it has not been explained before. What would you accomplish using a reference period, what are you comparing?

Line 109, "an increase of their magnitude", can you explain it clearly?

Line 110, "a transfer of the month of occurrence" should be explained clearly. Which month was before? How do you know? Can you give some examples?

Line 121, could you provide some examples of hourly intensity?

Line 127, can you provide some amount of the losses of Los Alcázares? It will help to give a context within the south-east Spanish peninsula.

Line 133, where was located the affected area regarding "Klaus"?

Table 1. When your refer to surface area occupied by buildings, is it related to the municipality surface? Please clarify it.

Lines 156-157 are not clear enough from my point of view. What do you mean with "infrastructures erected"? Public buildings? Roads? Bridges?

Lines 159-160Can you explain why the number of occupied land in San Javier is higher than in Los Alcázeres? Could be related to the municipality surface? Is one of them larger than the other?

Lines 163-164 what is the meaning of recognized buildings?

Line 171 some synonym of "vigour" could be find?

Line 213 are they rivers? Or ephemeral streams?

Section A and B could be 3.3.1 and 3.3.2?

Could you explain before how do you get the return periods?

Lines 244 to 256 should be rewritten an be clearer. The current version is confusing.

Lines 289 to 295 same as before.

Lines 314 to 315 not clear as well.

Regarding all the figures and tables of section 3, please check carefully the captions and make them clearer.

**Conclusion**

Line 374 can you explain what do you mean with "despite not having the social impact they currently deserve"? If you are studying them is because they do have an impact, at least that's the idea that someone can gather reading your manuscript.

Line 377 you wrote about "the meteorological irregularity that characterizes the Mediterranean climate..." I think such characteristics should be explained somewhere, before the conclusion. Please take into account that some readers could not be aware of the Mediterranean climate trends.

Line 378 "long periods of low water"? Does it means that the streams have no runoff recorded because it does not rain or can be other causes, such as irrigation measures that eliminate runoff. Please check the phrase and rewrite it.

Lines 388 to 397 "After analysing both point of view", where it has been analyzed? Rewrite it to make it clearer.

"Intense precipitation events …..considered as triggers" as is usual in Mediterranean ephemeral streams, something that should be explained, it is not only a common occurrence in your area of study.

"perpetration of these catastrophes" should be rewritten as well. As I pointed out before, the question is if a flood is truly a catastrophe?

Line 399 "San Javier is disproportionately increasing..." Still happens today? Even after the 2008 crisis that affected the building sector in Spain? And it is increasing compared to? The whole Spain? The region? Please clarify such affirmation.

Line 408 "not forgetting….Mediterranean climate" has been stated before, there is no need to explain it again.

Lines 411 415 should be rewritten again. From my point of view, what you are trying to convey is not clear enough.

In summary, this manuscript will definitely be of interest and therefore should, in my opinion, be published in NHESS. I am looking forward to it but, given the considerable number of comments, I suggest that the paper be reconsidered after **major revisions**.